# Risks of adverse perinatal and maternal outcomes among women with hypertensive disorders of pregnancy in southwestern Uganda

Henry Mark Lugobe[1]*, Rose Muhindo[2], Musa Kayondo[1], Ian Wilkinson[3], David Collins Agaba[1], Carmel McEniery[3], Samson Okello[2], Blair J. Wylie[4], Adeline A. Boatin[5]

1 Department of Obstetrics and Gynecology, Mbarara University of Science and Technology, Mbarara, Uganda, 2 Department of Internal Medicine, Mbarara University of Science and Technology, Mbarara, Uganda, 3 University of Cambridge, Cambridge, United Kingdom, 4 Division of Maternal-Fetal Medicine, Department of Obstetrics and Gynecology, Beth Israel Deaconess Medical Center, Harvard Medical School, Boston, Massachusetts, United States of America, 5 Department of Obstetrics and Gynecology, Massachusetts General Hospital, Harvard Medical School, Boston, MA, United States of America

* henrylugobe@must.ac.ug

**Data Availability Statement:** All relevant data are within the manuscript and its Supporting Information files

## Abstract

### Introduction

Hypertensive disorders of pregnancy (HDP) are a leading cause of global perinatal (fetal and neonatal) and maternal morbidity and mortality. We sought to describe HDP and determine the magnitude and risk factors for adverse perinatal and maternal outcomes among women with HDP in southwestern Uganda.

### Methods

We prospectively enrolled pregnant women admitted for delivery and diagnosed with HDP at a tertiary referral hospital in southwestern Uganda from January 2019 to November 2019, excluding women with pre-existing hypertension. The participants were observed and adverse perinatal and maternal outcomes were documented. We used multivariable logistic regression models to determine independent risk factors associated with adverse perinatal and maternal outcomes.

### Results

A total of 103 pregnant women with a new-onset HDP were enrolled. Almost all women, 93.2% (n = 96) had either pre-eclampsia with severe features or eclampsia. The majority, 58% (n = 60) of the participants had an adverse perinatal outcome (36.9% admitted to the neonatal intensive care unit (ICU), 20.3% stillbirths, and 1.1% neonatal deaths). Fewer participants, 19.4% (n = 20) had an adverse maternal outcome HELLP syndrome (7.8%), ICU admission (3%), and postpartum hemorrhage (3%). In adjusted analyses, gestational age of < 34 weeks at delivery and birth weight <2.5kg were independent risk factors for adverse

**Funding:** Research reported in this publication was supported by the Fogarty International Center and co-founding partners (NIH Common Fund, Office of Strategic Coordination, Office of the Director (OD/OSC/CF/NIH); Office of AIDS Research, Office of the Director (OAR/NIH); National Institute of Mental Health (NIMH/NIH); and National Institute of Neurological Disorders and Stroke (NINDS/NIH)) of the National Institutes of Health under Award Number D43TW010128 to HML. The content is solely the responsibility of the authors and does not necessarily represent the official views of the National Institutes of Health. The funders had no role in study design, data collection, and analysis, decision to publish, or preparation of the manuscript.

**Competing interests:** The authors have declared that no competing interests exist.

perinatal outcomes while referral from another health facility and eclampsia were independent risk factors for adverse maternal outcomes.

## Conclusion

Among women with HDP at our institution, majority had preeclampsia with severe symptoms or eclampsia and an unacceptably high rate of adverse perinatal and maternal outcomes; over a fifth of the mothers experiencing stillbirth. This calls for improved antenatal surveillance of women with HDP and in particular improved neonatal and maternal critical care expertise at delivering facilities. Earlier detection and referral, as well as improvement in initial management at lower level health units and on arrival at the referral site is imperative.

## Introduction

Hypertensive disorders of pregnancy (HDP) including gestational hypertension, preeclampsia, and eclampsia complicate 2–8% of all pregnancies globally [1]. In developing countries where the incidence of HDP is estimated to be seven times higher compared to developed countries, these disorders are a leading cause of maternal and neonatal mortality and morbidity [2]. At the Mbarara Regional Referral Hospital in Uganda, a tertiary care hospital caring for a largely rural and agrarian population, HDP is the third leading cause of maternal death after puerperal sepsis and obstetric hemorrhage [3]. Reducing and averting morbidity and mortality from HDP requires early detection, treatment with antihypertensive therapy, seizure prophylaxis and prompt delivery in severe cases [4–7].

To date, most studies examining maternal and perinatal outcomes in women with HDP from sub-Saharan Africa have been limited to retrospective and cross-sectional data [7–17]. Additionally most studies from sub-Saharan Africa are from institutions based in large urban and metropolitan areas, thus data is lacking on outcomes among women with HDP seeking care in more rural and agrarian settings [9, 12, 15, 18]. Existing data suggest an increased risk of maternal and perinatal adverse outcomes in women with HDP compared to women without HDP [15, 17, 19, 20]. Several demographic (age, education, rural domicile), medical (pre-existing hypertension, multiparity, gestational age at delivery) and management (timing of drug administration) characteristics have been identified as risk factors for adverse perinatal and maternal outcomes among women with HDP [9–11, 14, 15, 21–23]. However, it is unknown if similar rates of adverse outcomes and risk factors persist in a mostly rural population.

An improved understanding of the rate of adverse outcomes and characteristics associated with these outcomes, will provide the basis for health system improvement and the development of protocols to guide the identification of at-risk mothers and babies and implement interventions to improve their birth outcomes.

We sought to describe HDP in a prospective cohort of women delivering in rural Uganda and determine the magnitude and risk factors for adverse perinatal and maternal outcomes among women with HDP at Mbarara Regional Referral Hospital.

## Material and methods

### Study design and setting

We conducted a prospective cohort study of women with hypertensive disorders of pregnancy admitted for delivery at Mbarara Regional Referral Hospital (MRRH), in Southwestern

Uganda from January 2019 to November 2019. MRRH is a government funded public hospital that conducts approximately 9000 deliveries per year, with a caesarean delivery rate of 40%, maternal mortality rate of 261 per 100,000 live births and perinatal mortality rate of 33 per 1000 live birth according to the 2019 hospital records.

## Participants

Our study population included all pregnant women, including emancipated minors under the age of 18 years, at $\geq$ 20 weeks of gestation with new-onset hypertension in pregnancy diagnosed at admission. We defined hypertension as two blood pressure readings with either a systolic blood pressure $\geq$ 140 or diastolic $\geq$ 90 mmHg) measured 4 hours apart. Women reporting a history of hypertension prior to pregnancy or prior to 20 weeks of gestation diagnosed by a health care provider, or taking an antihypertensive medication prior to pregnancy were considered to have chronic hypertension and excluded from participation.

A prior study auditing vital sign assessment at MRRH found only 50% of women had blood pressure checked as part of routine clinical care on admission [24]. Thus, to capture women meeting the study eligibility criteria, all pregnant women presenting at the maternity ward of MRRH for admission, had a screening blood pressure performed by research staff on admission. Women with an elevated blood pressure at admission had a subsequent check 4 hours later. Women who were normotensive at admission and developed an elevated blood pressure at a later time point during their labor course or postpartum course were not included in the study. Women who met the inclusion criteria and consented to the study were enrolled at admission and followed up to 12 weeks postpartum. This study was part of a prospective cohort study to examine the magnitude of persistent hypertension at 12 weeks postpartum after HDP. Mothers were interviewed at admission, discharge, 6 weeks postpartum and 12 weeks postpartum by trained study staff.

## Variables and data sources

We classified enrolled women a priori as those with gestational hypertension, pre-eclampsia, pre-eclampsia with severe features and eclampsia at admission. Gestational hypertension was defined as new-onset hypertension without proteinuria. Pre-eclampsia was defined as new onset hypertension with proteinuria. The study team assessed for proteinuria (defined as $\geq$ 2 + protein) in all enrolled women at admission using a dipstick of a mid-stream urine sample. Severe features of pre-eclampsia included any of the following: blood pressure of $\geq$160 mmHg systolic or $\geq$110 mmHg diastolic, $\geq$ 3+ protein by dipstick, persistent epigastric pain, persistent headache, visual changes or elevated serum creatinine [4, 25]. A blood sample was drawn at recruitment for study purposes and analysed for renal (serum creatinine and urea) function. Liver function tests and complete blood count were abstracted from the chart if performed for clinical purposes. Women with grand mal seizures unrelated with other cerebral conditions who had signs and symptoms of preeclampsia were defined as having eclampsia [26].

The primary outcome measures were adverse perinatal and maternal outcomes occurring during the participant's hospital stay or within seven days of delivery, whichever came first and these were determined a priori. An adverse perinatal outcome was defined as a composite of one or more of the following: antepartum stillbirth (fetal death prior to the onset of labor), intrapartum stillbirth (fetal death occurring after the onset of labor and prior to delivery), admission to the neonatal intensive care unit (NICU) or neonatal death by discharge or within seven days, whichever came first. Adverse maternal outcome was defined as a composite of one or more of the following: hysterectomy, laparotomy (women delivered vaginally who later require abdominal exploration, or a post cesarean section mother who required re-operation

for abdominal exploration), primary postpartum hemorrhage (estimated blood loss of > 500mls following vaginal delivery or >1000mls following cesarean delivery within 24 hours after delivery as documented in the participant's medical forms), admission to Intensive Care Unit (ICU), stroke, HELLP (Hemolysis Elevated Liver enzymes and Low Platelets) syndrome and blood transfusion. All adverse outcomes were obtained by chart abstraction from the participant medical forms.

Other covariates of interest included socio-demographic characteristics, past medical history, obstetric history and obstetric care factors. This information was obtained using an interviewer administered questionnaire during participant enrollment and a second questionnaire administered at discharge. Socio-demographic data included maternal age, marital status, level of education and referral status (i.e. women referred in to MRRH from another health center). Medical history included HIV status (HIV positive or negative result done within 3 months), history of chronic kidney disease and history of diabetes mellitus. Pre-pregnancy body mass index (BMI) was not available, however, we measured weight and height at enrollment to the study, and calculated the BMI (weight in kilograms divided by height in meters squared) from this measurement. We then classified women as underweight ($<18.5kg/m^2$), normal weight ($18.5–24.9kg/m^2$), overweight ($25-29kg/m^2$) and obese $\geq30kg/m^2$) at data analysis. Obstetric data collected included parity, mode of delivery, gestational age at delivery (determined primarily using the last normal menstrual period (LNMP) or first trimester obstetric ultrasound scan if available and LNMP was unknown), history of hypertension in previous pregnancy and eclampsia (grand mal convulsions). Obstetric care factors included the time to delivery (calculated as admission date/time to date/time of delivery), administration of antenatal corticosteroids (for women at 28 to 33 weeks of gestation according to the MRRH protocols), magnesium sulfate and antihypertensive treatment, captured from chart abstraction of the participant medical forms. Study data were collected and managed using REDCap electronic data capture tools hosted at MUST Department of OB/GYN. REDCap (Research Electronic Data Capture) is a secure, web-based software platform designed to support data capture for research studies [27, 28].

## Sample size and power

The sample size was based on the primary cohort study examining persistent hypertension. For this sub-study, sample size and power were therefore not determined a priori.

## Data analysis

Maternal socio-demographic, medical and obstetric characteristics were presented in frequency tables stratified by type of hypertensive disorder. Univariate analysis for risk factors for adverse maternal and perinatal outcomes was performed using crude risk ratios. Factors with a p value of $\leq 0.2$ at univariate analysis were considered for inclusion in the adjusted analysis. Multiple logistic regression models were used to determine the independent risk factors with their corresponding 95% confidence intervals [29]. A p-value less than 0.05 was considered statistically significant. Data analysis was performed with Stata version 15. (Statacorp, College Station, TX, USA).

## Ethical consideration

The study procedures were approved by the Mbarara University Research Ethics Committee (07/09-18), Uganda National Council for Science and Technology (HS 2570) and Partners Healthcare Institutional Review Board (2019P001446). Participants provided written informed consent.

## Results

There were 9,096 deliveries at MRRH during the study period (January-November 2019). Of these, 155 women with hypertension in pregnancy upon admission to the maternity ward were screened for eligibility. Of these, 6 were excluded due to pre-existing hypertension before pregnancy, and 38 declined consent. Thus, a total 103 women with new-onset hypertension during pregnancy were enrolled. Of these 103 women, 4.9% (n = 5) had gestational hypertension, 1.9% (n = 2) had preeclampsia with mild features, 71.8% (n = 74) had pre-eclampsia with severe features and 21.4% (n = 22) had eclampsia.

Participant characteristics are shown in Table 1. The mean age was 27 years (SD±6). Most participants were multigravida 65% (n = 67), referred from other health facilities 63.1% (n = 65) and delivered preterm at gestational age <37 weeks 61.3% (n = 57). Over a quarter of participants were obese (27.6%, n = 27). Most women, 61.8% (n = 63) were delivered by cesarean section and 42.2% (n = 43) within 24 hours of admission. The majority of babies had either a low birth weight 37.1% (n = 36) or a very low birth weight 20.6% (n = 20). Almost all the participants 96.1% (n = 99) received anti-hypertensive medication and 92.6% (n = 87) received magnesium sulfate. Less than half, 44.7% (n = 42) received a complete dose of magnesium sulfate, only 31% (n = 9) of the women at gestational age < 34 weeks received antenatal corticosteroids. Data on age 1% (n = 1), body mass index 4.8% (n = 5), gestational age at delivery 9.7% (n = 10), eclampsia 3.9% (n = 4), mode of delivery 1% (n = 1), time to delivery 1% (n = 1), and magnesium sulfate administration 8.7% (n = 9) were missing.

Adverse perinatal and maternal outcomes are reported in Table 2. Overall, 58% (n = 60) participants experienced one or more adverse perinatal outcome and 19.4% (n = 20) one or more adverse maternal outcome. Adverse perinatal outcomes included admission to the NICU 36.9% (n = 38), stillbirth 20.3% (n = 21) and neonatal death before discharge 1.1% (n = 1). Of the 21 stillbirths, 76.2% (n = 16) were antepartum stillbirths and 23.8% (n = 5) were intrapartum stillbirths. Adverse maternal outcome included HELLP syndrome 7.8% (n = 8), blood transfusion 2.9% (n = 3), ICU admission 2.9% (n = 3) and postpartum hemorrhage 2.9% (n = 3).

In an adjusted analysis controlling for history of HDP in prior pregnancy, administration of magnesium sulfate and gravidity, the independent risk factors for adverse perinatal outcomes were gestational age at delivery <34 (aRR 1.6; 95% CI, 1.2–2.3; p<0.01) and birth weight <2.5kg (aRR 1.4; 95% CI, 1.1–2.1; p = 0.02) as shown in Table 3.

In an adjusted analysis controlling for level of education, mode of delivery and magnesium sulfate administration, the independent risk factors for adverse maternal outcomes were referral from another facility (aRR 3.9; 95% CI, 1.1–13.8; p< 0.01) and eclampsia (aRR 3.7; 95% CI, 1.6–8.4; p = 0.01) as shown in Table 4.

## Discussion

In this prospective cohort study of women with hypertensive disorders of pregnancy we found a high rate of severe HDP, and high rates of both perinatal and maternal adverse outcomes, with almost two-third of babies having an adverse perinatal outcome and a fifth of women experiencing an adverse maternal outcome. Preterm delivery and low birth weight were independent significant risk factors for adverse perinatal outcomes as expected. Referral from another facility and eclampsia were independent and significant risk factors for adverse maternal outcomes.

The majority of women in this cohort had either pre-eclampsia with severe disease or eclampsia; together accounting for 93.2% of the cohort. The high rates of adverse perinatal and maternal outcomes reflects this severity and are comparable to findings in other SSA settings.

**Table 1. Participant characteristics.**

| Participant Characteristics n = 103 | | n (%) |
|---|---|---|
| Age n = 102 | <35 | 85 (83.3) |
| | ≥35 | 17 (16.7) |
| Marital Status | Not Married | 7 (6.8) |
| | Married | 96 (93.2) |
| Level of Education | Primary and below | 50 (48.5) |
| | Secondary and above | 53 (51.5) |
| Referral from another facility | Not Referred | 38 (36.9) |
| | Referred | 65 (63.1) |
| Gravidity | Primgravida | 36 (35) |
| | Multigravida | 67 (65) |
| History of HDP in a prior pregnancy | Yes | 13(12.6) |
| | No | 90 (87.4) |
| Body Mass Index n = 98 | Underweight | 1 (1) |
| | Normal | 30 (30.6) |
| | Over weight | 40 (40.8) |
| | Obese | 27 (27.6) |
| Gestational Age at delivery n = 93 | <28 | 6 (6.4) |
| | 28–33 | 29 (31.2) |
| | 34–36 | 22 (23.7) |
| | ≥37 | 36 (38.7) |
| HIV status | Positive | 7 (6.8) |
| | Negative | 96 (93.2) |
| Eclampsia n = 99 | No | 77 (77.8) |
| | Yes | 22 (22.2) |
| Mode of Delivery n = 102 | Vaginal delivery | 39 (38.2) |
| | Cesarean section | 63 (61.8) |
| Time to delivery n = 102 | Within 24 hours | 43 (42.2) |
| | > 24 hours | 59 (57.8) |
| Steroid administration | Received | 9 (31) |
| | Not Received | 20 (69) |
| Anti-Hypertensive medicine | Received | 99 (96.1) |
| | Not Received | 4 (3.9) |
| Magnesium Sulfate administration n = 94 | Complete dose | 42 (44.7) |
| | Loading dose only | 19 (20.2) |
| | Incomplete dose | 26 (27.7) |
| | Not Received | 7 (7.4) |

Reported stillbirth rates among women with HDP in SSA range from 6.8%-22.6%, NICU admissions from 24.7% - 28.8% and neonatal deaths from 3% - 14.1% [7, 9, 11–13, 15, 16, 30]. In cohorts of women with HDP in SSA, HELLP syndrome is reported in 0.8%-13% of women, PPH in 5.9%-7% and ICU admission in 5.7%-29.3% [13–15, 17, 18, 26, 31]. Our findings on the rate of NICU admission and neonatal deaths fall in the lower end of the range of those reported these studies. However, we found a rate of stillbirth that falls on the high end of the reported range and includes a high proportion (over 70%) of antepartum stillbirths. HELLP syndrome was the commonest adverse maternal outcome and may be under reported as not all patients in this cohort had lab evaluation beyond those done for study purposes. This shows the need to have laboratory services at health facilities able to do full blood count and liver

**Table 2. Frequency and distribution of adverse perinatal and maternal outcomes.**

| Adverse perinatal outcome | n (%) |
|---|---|
| Intrapartum stillbirth | 5 (4.8) |
| Antepartum stillbirth | 16 (15.5) |
| Admission to Neonatal Intensive Care Unit | 38 (36.9) |
| Death of infant before discharge | 1 (1.1) |
| *One or more adverse perinatal outcome* | 60 (58) |
| **Adverse maternal outcomes** | **n (%)** |
| Hysterectomy | 1 (1.0) |
| Laparotomy | 1 (1.0) |
| Blood transfusion | 3 (2.9) |
| Admission to Intensive Care Unit | 3 (2.9) |
| Postpartum hemorrhage | 3 (2.9) |
| Stroke | 1 (1.0) |
| HELLP syndrome | 8 (7.8) |
| *One or more adverse maternal outcome* | 20 (19.4) |

function tests. This will help in early identification of these women with evidence of organ dysfunction in order to institute the appropriate management.

These findings, along with the risk factors identified including gestational age at delivery, low birth weight, eclampsia and referral from another facility have several implications. This is because these factors suggest a possible delay in early detection of HDP, prompt treatment with antihypertensive therapy, timely administration of seizure prophylaxis and prompt delivery which measures have been shown to reduce and avert morbidity and mortality. Preterm delivery in many cases is due to the need for delivery to enable definitive management for HDP. Low birth weight as an independent risk factor likely represents undiagnosed intrauterine fetal growth restriction. However, the need for early delivery, and evidence for fetal growth restriction emphasize the severity of disease in this cohort and point to the need for strengthening antenatal fetal surveillance and the monitoring of mothers with hypertensive disorders of pregnancy, as well as the need for prevention strategies such as the use of aspirin prophylaxis in appropriate candidates [32]. Secondly, once delivered it is clear that babies born to women with HDP are in need of prompt and specialized care. Our findings highlight the need for functional NICUs, trained NICU nurses and neonatologists and protocols for neonatal resuscitation to improve the outcomes of the babies delivered prematurely and with low birth weight.

Furthermore, women with eclampsia and those who were referred from another facility had a significantly increased risk for adverse maternal outcomes. Mothers with eclampsia may have neurological dysfunction, metabolic coma, stroke, uncontrollable fits and eclamptic encephalopathy [26]. The severity of disease on admission with a high rate of severe features and high rate of eclampsia emphasizes gaps in early identification and treatment. Delays in diagnosis and management due to referral from one facility to another are well described in the obstetric literature [33, 34]. Transfer from one facility to another introduces several time points where gaps in care may occur: at the facility initiating the referral, en route, and upon arrival at the receiving facility. At each stage, inadequate numbers of trained staff, equipment, medications and systems to identify at-risk women compound the potential for women to develop complications.

Although we demonstrate a high rate of anti-hypertensive use in this population, we noted less than half of the women received appropriate dosing of magnesium sulfate, and even fewer

**Table 3. Risk factors for adverse perinatal outcomes.**

| Characteristic | | Adverse fetal outcome n = 60 | Crude Risk Ratio | p value | Adjusted Risk Ratio | p value |
|---|---|---|---|---|---|---|
| Age category | <35 | 48 (80.0) | ref | | | |
| | ≥35 | 12 (20.0) | 1.3 (0.9,1.8) | 0.28 | - | - |
| Marital Status | Single | 5 (8.3) | ref | | | |
| | Married | 55 (91.7) | 0.8 (0.5,1.3) | 0.46 | - | - |
| Level of Education | Primary and below | 30 (50.0) | ref | | | |
| | Secondary and above | 30 (50.0) | 0.9 (0.7,1.3) | 0.73 | - | - |
| Referred from another health center | Not Referred | 21 (35.0) | ref | | | |
| | Referred | 39 (65.0) | 1.1 (0.8,1.5) | 0.64 | - | - |
| HIV Status | Negative | 56 (93.3) | ref | | | |
| | Positive | 4 (6.7) | 1.0 (0.5,1.9) | 0.95 | - | - |
| Gravidity | Primgravida | 14 (23.3) | ref | | | |
| | Multigravida | 46 (76.7) | 1.8(1.1,2.7) | <0.01 | 1.3 (0.9,1.8) | 0.09 |
| History of HDP in prior pregnancy | No | 49 (81.7) | ref | | | |
| | Yes | 11 (18.3) | 1.6 (1.2,2.1) | 0.04 | 1.2(0.7,2.0) | 0.40 |
| Mode of Delivery | Vaginal delivery | 26 (43.3) | ref | | | |
| | C-section | 34 (56.7) | 0.8 (0.6,1.1) | 0.21 | - | |
| Gestational Age at Delivery | ≥34 | 22 (40.0) | ref | | | |
| | < 34 | 33 (60.0) | 2.5 (1.8,3.5) | <0.01 | 1.6(1.2,2.3) | <0.01 |
| BMI at admission | Not Obese (<30) | 40 (71.4) | ref | | | |
| | Obese (≥30) | 16 (28.6) | 1.1 (0.7,1.5) | 0.79 | - | - |
| Eclampsia | No | 43 (74.1) | ref | | | |
| | Yes | 15 (25.9) | 1.2(0.9,1.7) | 0.33 | - | - |
| Birth weight | ≥ 2.5kg | 13 (23.2) | ref | | | |
| | <2.5kg | 43 (76.8) | 2.4 (1.5,3.9) | <0.01 | 1.4(1.1,2.1) | 0.02 |
| Time to Delivery | Within 24 hours | 22 (36.7) | ref | | | |
| | > 24 hours | 38 (63.3) | 1.2 (0.9,1.8) | 0.21 | - | - |
| Anti- Hypertensive | Given | 59 | ref | | | |
| | Not Given | 1 | 0.4 (0.1,2.3) | 0.17 | - | - |
| Steroid administration | Given | 9 | ref | | | |
| | Not Given | 18 | 0.9 (0.8,1.0) | 0.32 | - | - |
| Magnesium Sulfate administration | Given | 32 | ref | | | |
| | Not Given | 23 | 0.6 (0.4,0.8) | 0.01 | 0.8 (0.6,1.0) | 0.07 |

Adjusted for history of HDP in prior pregnancy, administration of magnesium sulfate and gravidity.

antenatal steroids for fetal lung maturity with a trend towards the most severe cases, i.e. those receiving treatments had a non-significant increase in the risk of adverse outcomes. These findings highlight a need to develop training protocols and ensure drug availability that can extend prophylactic measures to more women to prevent adverse outcomes. Given the severity noted at admission to this tertiary care center it is clear that such management needs to occur earlier in the referral chain. Thus, a focus on ensuring training protocols and system reforms occur at lower level health care will be essential to improve outcomes among women with HDP.

Our study has some limitations. The study was conducted in a single regional referral hospital in southwestern Uganda and the findings may not be generalizable to all the other regional referral hospitals in Uganda, or other levels of facilities. As a referral hospital, our cohort may be skewed towards women with more advanced or severe disease. Additionally, we

**Table 4. Risk factors for adverse maternal outcomes.**

| Characteristic | | Adverse maternal outcomes | Crude Risk Ratio | p value | Adjusted Risk Ratio | p value |
|---|---|---|---|---|---|---|
| | | Yes = 20 | | | | |
| Age category | <35 | 18 (90.0) | ref | | | |
| | ≥35 | 2 (10.0) | 0.6 (0.1,2.2) | 0.37 | - | - |
| Marital Status | Single | 1 (5.0) | ref | | | |
| | Married | 19 (95.0) | 1.4 (0.2,8.9) | 0.72 | - | - |
| Level of Education | Primary and below | 7 (35.0) | ref | | | |
| | Secondary and above | 13 (65.0) | 1.8 (0.8,4.0) | 0.18 | 1.7 (0.8,3.6) | 0.14 |
| Referred from another health center | Not Referred | 4 (20.0) | ref | | | |
| | Referred | 16 (80.0) | 2.3 (0.8,6.5) | 0.08 | 3.9 (1.1,13.8) | <0.01 |
| HIV Status | Negative | 19 (95.0) | ref | | | |
| | Positive | 1 (5.0) | 0.7 (0.1,4.6) | 0.72 | - | - |
| Gravidity | Primgravida | 8 (40.0) | ref | | | |
| | Multigravida | 12 (60.0) | 0.8 (0.4,1.8) | 0.60 | - | - |
| History of HDP in prior Pregnancy | No | 17 (85.0) | ref | | | |
| | Yes | 3 (15.0) | 1.2 (0.4, 3.6) | 0.72 | - | - |
| Mode of Delivery | Vaginal delivery | 5 (25.0) | ref | | | |
| | C-section | 15 (75.0) | 1.8 (0.7,4.7) | 0.17 | 1.8 (0.7,4.8) | 0.26 |
| Body Mass Index | Not Obese (<30) | 14 (77.8) | ref | | | |
| | Obese (≥30) | 4 (22.2) | 0.8 (0.3,2.1) | 0.58 | - | |
| Eclampsia | No | 8 (44.4) | ref | | | |
| | Yes | 10 (55.6 | 4.4 (1.9,9.7) | <0.01 | 3.7 (1.6,8.4) | 0.01 |
| Time to Delivery | Within 24 hours | 11 (57.9) | ref | | | |
| | > 24 hours | 8 (42.1) | 0.5 (0.2,1.2) | 0.11 | | |
| Anti- Hypertensive | Given | 20 | | | | |
| | Not Given | 0 | 0 | 0.32 | | |
| Steroid administration | Given | 3 | | | | |
| | Not Given | 2 | 0.3 (0.1,1.4) | 0.12 | | |
| Magnesium Sulfate administration | Given | 14 | | | | |
| | Not Given | 5 | 0.3 (0.1,0.7) | 0.01 | 0.4 (0.2,1.1) | 0.05 |

Adjusted for level of education, mode of delivery and magnesium sulfate administration.

were only able to obtain adverse outcomes that occurred while in the hospital, therefore we would not have captured any additional adverse outcomes that occurred after discharge and in the community or another facility. Our estimates may thus underestimate the true burden in this cohort of women. In our study HELLP syndrome was reported documentation available in the participants' medical records. Thus liver function tests and platelet counts were only available if done for clinical purposes. As these lab tests are not routinely available, this might have led to under estimation of participants with HELLP syndrome for those who did not do the tests. Lastly, we did not enroll women who were normotensive at admission and developed hypertension during the course of their labour and in the postpartum period, so we likely underestimated the burden of hypertension at this facility.

## Conclusion

Most women in this cohort had an adverse perinatal outcome, with over a fifth experiencing a stillbirth. Given stagnation in reduction of global stillbirth and neonatal mortality rates, this study identifies women with preeclampsia as target group. Similarly, given high rates of

specialized neonatal care needed and risk factors of prematurity and low birth weight, strategies are needed to identify women at risk and triage delivery to facilities with neonatal expertise. There is need to have well equipped and functional NICU, NICU nurses and neonatologists. Quality improvement strategies also need to be put in place to target the referral pathways and immediate critical care and stabilization of women with hypertensive disorders of pregnancy to prevent eclampsia and improve the outcomes of these mothers.

## Supporting information

**S1 Dataset.**
(DTA)

**S1 Questionnaire.**
(DOCX)

## Acknowledgments

The authors are grateful to the research assistants Ms. Patience Naiga, Ms. Florida Tusiimiraho, Ms. Daphine Kibanda, Dr. Ruth Grace Kakoba and Dr. Twesigomwe Godfrey. We acknowledge the staff at the maternity ward of Mbarara Regional Referral Hospital, Mbarara University of Science and Technology and all the study participants. Special thanks to professor Celestino Obua and all the members of the training advisory committee of Mbarara University Research Training Initiative for the guidance offered during this study.

## Author Contributions

**Conceptualization:** Henry Mark Lugobe, Rose Muhindo, Musa Kayondo, Adeline A. Boatin.

**Data curation:** Henry Mark Lugobe, David Collins Agaba, Carmel McEniery, Samson Okello.

**Formal analysis:** Henry Mark Lugobe, Samson Okello, Blair J. Wylie, Adeline A. Boatin.

**Funding acquisition:** Henry Mark Lugobe.

**Investigation:** Henry Mark Lugobe, Adeline A. Boatin.

**Methodology:** Henry Mark Lugobe, Rose Muhindo, Musa Kayondo, Ian Wilkinson, David Collins Agaba, Carmel McEniery, Blair J. Wylie, Adeline A. Boatin.

**Writing – original draft:** Henry Mark Lugobe, Rose Muhindo, Musa Kayondo, Ian Wilkinson, David Collins Agaba, Carmel McEniery, Samson Okello, Blair J. Wylie, Adeline A. Boatin.

**Writing – review & editing:** Henry Mark Lugobe, Rose Muhindo, Musa Kayondo, Ian Wilkinson, David Collins Agaba, Carmel McEniery, Samson Okello, Blair J. Wylie, Adeline A. Boatin.

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
