## [Decision Letter · Decision Letter 0]

8 Sep 2020

PONE-D-20-20782

Risks of adverse perinatal and maternal outcomes among women with hypertensive disorders of pregnancy in southwestern Uganda

PLOS ONE

Dear Dr. Lugobe,

Thank you for submitting your manuscript to PLOS ONE. After careful consideration, we feel that it has merit but does not fully meet PLOS ONE’s publication criteria as it currently stands. Therefore, we invite you to submit a revised version of the manuscript that addresses the points raised during the review process.

Please address the issues raised, specifically:

Clarify which predictors are controlled for in the adjusted risk ratios reported in Tables 3 and 4Clarify your intention in regard to the prevalence of hypertensive disorders in your facility (Question 2, Reviewer 1). This also relates to the question raised about the proportion of women without history of pre-pregnancy hypertension that were found to be hypertensive on admission and so included in the study and the inclusion of HELLP syndrome as an outcome variable.Consider my suggestion for the discussion to more explicitly link back to the list of issues known to reduce morbidity and mortality in referred to the introduction: early detection, treatment with anti-hypertensive therapy, seizure prophylaxis and prompt delivery in severe cases.  ==============================Please submit your revised manuscript by Oct 23 2020 11:59PM. If you will need more time than this to complete your revisions, please reply to this message or contact the journal office at plosone@plos.org. Please include the following items when submitting your revised manuscript:A rebuttal letter that responds to each point raised by the academic editor and reviewer(s). You should upload this letter as a separate file labeled 'Response to Reviewers'.A marked-up copy of your manuscript that highlights changes made to the original version. You should upload this as a separate file labeled 'Revised Manuscript with Track Changes'.An unmarked version of your revised paper without tracked changes. You should upload this as a separate file labeled 'Manuscript'.

We look forward to receiving your revised manuscript.

Kind regards,

Dell Horey

Academic Editor

PLOS ONE

Journal Requirements:

2. Please provide additional details regarding participant consent. In your Methods section, please ensure you have also stated whether your study included minors, whether you obtained consent from parents or guardians of any minors included in the study, or whether the research ethics committee or IRB specifically waived the need for their consent.

3. Please include additional information regarding the survey or questionnaire used in the study and ensure that you have provided sufficient details that others could replicate the analyses. For instance, if you developed a questionnaire as part of this study and it is not under a copyright more restrictive than CC-BY, please include a copy, in both the original language and English, as Supporting Information."

4. Please provide a sample size and power calculation in the Methods, or discuss the reasons for not performing one before study initiation."

5. Please include an explanation for and description of the missing data found in Table 1

Additional Editor Comments (if provided):

My apologies for the delay in returning this feedback to you.

You manuscript has been reviewed by one reviewer and by myself (as the second reviewer).

Reviewers' comments:

Reviewer's Responses to Questions

**Comments to the Author**

1. Is the manuscript technically sound, and do the data support the conclusions?

Reviewer #1: Yes

Academic Editor: Yes

2. Has the statistical analysis been performed appropriately and rigorously? 

Reviewer #1: Yes

Academic Editor: Yes

3. Have the authors made all data underlying the findings in their manuscript fully available?

Reviewer #1: Yes

Academic Editor: Yes

4. Is the manuscript presented in an intelligible fashion and written in standard English?

Reviewer #1: Yes

Academic Editor: Yes

5. Review Comments to the Author

Reviewer #1: The authors present a prospective cohort study of women with hypertensive disorders of pregnancy who were managed as inpatients in a rural referral hospital in Uganda. The objective of the study was to determine the prevalence of fetal/neonatal and maternal morbidity among women with hypertensive disorders at their center and then to identity predictors for these adverse outcomes. The authors identity a high rate of severe hypertensive disorders of pregnancy in their cohort as well as high rates of both adverse fetal/neonatal, as well as adverse maternal outcomes. They argue for education efforts in the region to help providers with early recognition of hypertensive disorders of pregnancy and initiation of appropriate management in women of rural Uganda.

Questions or comments for the authors.

1. Statistical analysis. Among women with hypertensive disorders of pregnancy, the two primary outcomes of interest were an adverse composite perinatal (fetal/neonatal) outcome and an adverse composite maternal outcome. Independent predictors to be included in the multivariable logistic regression model were chosen based on a univariate level of significance of less than 0.2. The right-hand columns in Tables 3 and 4 report the adjusted risk ratios for the predictors that were included in the model. For each of these adjusted risk ratios, is the reported point estimate given while controlling for all other listed variables in the table with univariate p-values of less than 0.2? Please clarify in the text (lines 183-185 and lines 188-190) which predictors are being controlled for in each of the reported adjusted risk ratios.

2. The conclusion statement in the abstract states, “in this cohort, we found a high rate of severe HDP and an unacceptably high rate of adverse perinatal and maternal outcomes…”. How was ‘severe HDP’ of pregnancy defined? The overall prevalence of hypertensive disorders of pregnancy in your cohort seems relatively low (149/9096 = 1.6%). I agree that among women diagnosed at your facility with eclampsia or preeclampsia with severe features, these two groups made up a large proportion of all women with hypertensive disorders of pregnancy. If this is your intention with the statement in the conclusion, maybe provide a clarifying statement that states, among women with hypertensive disorders of pregnancy at our institution, a large proportion of them had eclampsia or preeclampsia with severe features.

3. HELLP syndrome was included as an outcome variable within the adverse maternal outcome composite, rather than as a diagnosis of a hypertensive disorders that made women eligible to be included in the cohort. I guess treating HELLP syndrome as an outcome seems reasonable, but I wonder if consideration should be made for not including it as an outcome but just including it as an inclusion criteria for eligibility in the cohort. I’m not sure which approach would be more valid.

Academic Editor: This is a potentially useful study that requires some minor revisions.

Prospective cohort study of pregnant women in rural Uganda to describe hypertensive disorders of pregnancy, and determine magnitude and risk factors for adverse perinatal and maternal outcomes. The abstract refers to the last of these aims and this is in the method. It would be helpful to have a clear aim in the abstract.

The introduction to the paper includes the statement that “Reducing and averting morbidity and mortality from HDP requires early detection, treatment with antihypertensive therapy, seizure prophylaxis and prompt delivery in severe cases” (lines 38-40, p 3). This statement would be a useful framework to return to in the discussion, to show how care compared with what is known to be effective. This information is there but the reader has to look for it. It would be useful for the authors to explicitly do this.

There are three classifications reported in the paper:

- degree of condition (Line 90, p 6)

- adverse perinatal outcome (line 105, p 6 to line 108, p 7)

- weight(line 127, p 7)

It is not clear whether these classifications determined a priori or post hoc. Could this information be included please? It is also unclear how many women presenting at the hospital had their blood pressure checked (that is proportion of women without history of pre-pregnancy hypertension were included.)

I question whether is appropriate to describe 42.2% as “almost half” (line 166,, p 9).

Minor typographical and and copy-editing issues

(line 86, p 6) missing word - “of” at end of line “should be magnitude of persistent hypertension”

(Line 90, p 6) – why described as “sub-classified “ surely this is just “classified”.

(line 167 p 9) should be “37.1% (n=36) or 20.6%”

Tables

(lines 168-170) there should be some indication that the authors are referring to incomplete data. Suggest that the total number that had any Magnesium Sulphate should be reported.

Table 1 p9-10 – add n value for all characteristics where reported data is <103

6. PLOS authors have the option to publish the peer review history of their article (what does this mean?). If published, this will include your full peer review and any attached files.

Reviewer #1: No

Reviewer #2: No

---

## [Author Response · Author response to Decision Letter 0]

19 Sep 2020

Response: The manuscript has been reformatted to be in accordance with the PLOS ONE style 

2. Please provide additional details regarding participant consent. In your Methods section, please ensure you have also stated whether your study included minors, whether you obtained consent from parents or guardians of any minors included in the study, or whether the research ethics committee or IRB specifically waived the need for their consent.

Response: Pregnant participants below the age of 18 years are considered emancipated minors in Uganda and with IRB approval permitted to give their own written consent to the study. Therefore written consent was obtained from all the study participants including those under 18 without the requirement of consent from parents or minors. This has been clarified in the methods on Page 5, Line 72-73. 

“ Our study population included all pregnant women, including emancipated minors under the age of 18, at ≥ 20 weeks of gestation with new-onset hypertension in pregnancy diagnosed at admission.”

3. Please include additional information regarding the survey or questionnaire used in the study and ensure that you have provided sufficient details that others could replicate the analyses. For instance, if you developed a questionnaire as part of this study and it is not under a copyright more restrictive than CC-BY, please include a copy, in both the original language and English, as Supporting Information."

Response: The questionnaire used to collect data was developed for the purposes of this study and has now been has been shared as supporting information as shown in line 290 at the end of the manuscript.

“S1 Questionnaire”

 4. Please provide a sample size and power calculation in the Methods, or discuss the reasons for not performing one before study initiation."

The data used was from a cohort study to understand the burden of persistent hypertension postpartum. The study had the power and sample size calculation performed. For this paper, we described the data from all the enrolled participants as they had been consecutively enrolled and therefore we did not perform a sample size and power calculation. This is shown line 143-145 in the methods section.

“Sample size and power

The sample size was based on the primary cohort study examing persistent hypertension. For this sub-study, sample size and power were therefore not determined a priori.”

 5. Please include an explanation for and description of the missing data found in Table 1

The n for all the variables with missing data has been included in Table 1. An account of the variables with missing data has been included in the results section line 177-180. 

“Data on age 1% (n=1), body mass index 4.8% (n=5), gestational age at delivery 9.7% (n=10), eclampsia 3.9% (n=4), mode of delivery 1% (n=1), time to delivery 1% (n=1), and magnesium sulfate administration 8.7% (n=9) were missing.” 

Captions for supporting information files have been included line 288-291.

“Supporting information

S1 Dataset

S1 Questionnaire” 

Reviewers Comments:

1. Statistical analysis. Among women with hypertensive disorders of pregnancy, the two primary outcomes of interest were an adverse composite perinatal (fetal/neonatal) outcome and an adverse composite maternal outcome. Independent predictors to be included in the multivariable logistic regression model were chosen based on a univariate level of significance of less than 0.2. The right-hand columns in Tables 3 and 4 report the adjusted risk ratios for the predictors that were included in the model. For each of these adjusted risk ratios, is the reported point estimate given while controlling for all other listed variables in the table with univariate p-values of less than 0.2? Please clarify in the text (lines 183-185 and lines 188-190) which predictors are being controlled for in each of the reported adjusted risk ratios.

Response: The predictors that are being controlled for in each of the reported adjusted risk ratios have been included in the text in line 190-193 

“In an adjusted analysis controlling for history of HDP in prior pregnancy, administration of magnesium sulfate and gravidity, the independent risk factors for adverse perinatal outcomes were gestational age at delivery <34 (aRR 1.6; 95% CI, 1.2-2.3; p<0.01) and birth weight <2.5kg (aRR 1.4; 95% CI, 1.1-2.1; p=0.02) as shown in Table 3.”

and in line 195-198

“In an adjusted analysis controlling for level of education, mode of delivery and magnesium sulfate administration, the independent risk factors for adverse maternal outcomes were referral from another facility (aRR 3.9; 95% CI, 1.1-13.8; p< 0.01) and eclampsia (aRR 3.7; 95% CI, 1.6-8.4; p=0.01) as shown in Table 4.”

2. The conclusion statement in the abstract states, “in this cohort, we found a high rate of severe HDP and an unacceptably high rate of adverse perinatal and maternal outcomes…”. How was ‘severe HDP’ of pregnancy defined? The overall prevalence of hypertensive disorders of pregnancy in your cohort seems relatively low (149/9096 = 1.6%). I agree that among women diagnosed at your facility with eclampsia or preeclampsia with severe features, these two groups made up a large proportion of all women with hypertensive disorders of pregnancy. If this is your intention with the statement in the conclusion, maybe provide a clarifying statement that states, among women with hypertensive disorders of pregnancy at our institution, a large proportion of them had eclampsia or preeclampsia with severe features.

Response: We agree with the revieviwer this could be further clarified and we have modified this statement in line 25-26 to: 

“Among women with HDP at our institution, majority had preeclampsia with severe symptoms or eclampsia and an unacceptably high rate of adverse perinatal and maternal outcomes, with over a fifth of the mothers experiencing a stillbirth.”

3. HELLP syndrome was included as an outcome variable within the adverse maternal outcome composite, rather than as a diagnosis of a hypertensive disorders that made women eligible to be included in the cohort. I guess treating HELLP syndrome as an outcome seems reasonable, but I wonder if consideration should be made for not including it as an outcome but just including it as an inclusion criteria for eligibility in the cohort. I’m not sure which approach would be more valid.

Response: In our study we included pregnant women with hypertension. We considered HELLP syndrome as an adverse outcome in line with how this was reported in other similar studies.

https://www.sciencedirect.com/science/article/abs/pii/S0020729214003725

https://journals.plos.org/plosone/article?id=10.1371/journal.pone.0213240

Prospective cohort study of pregnant women in rural Uganda to describe hypertensive disorders of pregnancy, and determine magnitude and risk factors for adverse perinatal and maternal outcomes. The abstract refers to the last of these aims and this is in the method. It would be helpful to have a clear aim in the abstract.

Response: We have modified the abstract to include a clear aim of the study in the introduction section of the abstract line 4-6.

“We sought to describe HDP and determine the magnitude and risk factors for adverse perinatal and maternal outcomes among women with HDP in southwestern Uganda.”

The introduction to the paper includes the statement that “Reducing and averting morbidity and mortality from HDP requires early detection, treatment with antihypertensive therapy, seizure prophylaxis and prompt delivery in severe cases” (lines 38-40, p 3). This statement would be a useful framework to return to in the discussion, to show how care compared with what is known to be effective. This information is there but the reader has to look for it. It would be useful for the authors to explicitly do this.

Response: A comparison between the factors identified in our study and the care known to be effective, has been made in line 227-230.

“These findings, along with the risk factors identified including gestational age at delivery, low birth weight, eclampsia and referral from another facility have several implications. This is because these factors suggest a possible delay in early detection of HDP, prompt treatment with antihypertensive therapy, timely administration of seizure prophylaxis and prompt delivery which measures have been shown to reduce and avert morbidity and mortality.” 

There are three classifications reported in the paper:

- degree of condition (Line 90, p 6)

- adverse perinatal outcome (line 105, p 6 to line 108, p 7)

- weight(line 127, p 7)

It is not clear whether these classifications determined a priori or post hoc. Could this information be included please? 

Response: The degree of the HDP was determined a priori and this has been clarified in line 92-93. 

“We classified enrolled women a priori as those with gestational hypertension, pre-eclampsia, pre-eclampsia with severe features and eclampsia at admission.”

The primary outcome of adverse perinatal and maternal outcomes was determined a priori and this has been clarified in line 105-107.

“The primary outcome measures were adverse perinatal and maternal outcomes occurring during the participant’s hospital stay or within seven days of delivery, whichever came first and these were determined a priori.”

Weight and height were measured on admission for delivery and BMI classification i.e. obese, underweight and normal weight were determined post hoc during data analysis. This has been clarified on line 130-131

“We then classified women as underweight (<18.5kg/m2), normal weight (18.5-24.9kg/m2), overweight (25-29kg/m2) and obese ≥30kg/m2) at data analysis.”

It is also unclear how many women presenting at the hospital had their blood pressure checked (that is proportion of women without history of pre-pregnancy hypertension were included.)

Response: Clarification has been made in line 80-82 that all women presenting for admission had their blood pressure checked. From the exclusion criteria, women who had a history of hypertension pre-pregnancy were excluded from the study.

“Thus, to capture women meeting the study eligibility criteria, all pregnant women presenting at the maternity ward of MRRH for admission, had a screening blood pressure performed by research staff on admission.”

I question whether is appropriate to describe 42.2% as “almost half” (line 166,, p 9).

Response: The phrase “almost half” has been deleted in line 173.

Minor typographical and and copy-editing issues

(line 86, p 6) missing word - “of” at end of line “should be magnitude of persistent hypertension”

Response: The word “of” has been added in line 88.

(Line 90, p 6) – why described as “sub-classified “ surely this is just “classified”.

Response: The word has been changed from sub-classified to classified in line 92.

(line 167 p 9) should be “37.1% (n=36) or 20.6%”

Response: The statement has been re-written in line 173-174 

“The majority of babies had either a low birth weight 37.1% (n=36) or a very low birth weight 20.6% (n=20).”

Tables

(lines 168-170) there should be some indication that the authors are referring to incomplete data. 

Response: An account of the variables with missing data has been included in the results section line 177-180. 

“Data on age 1% (n=1), body mass index 4.8% (n=5), gestational age at delivery 9.7% (n=10), eclampsia 3.9% (n=4), mode of delivery 1% (n=1), time to delivery 1% (n=1), and magnesium sulfate administration 8.7% (n=9) were missing.” 

Suggest that the total number that had any Magnesium Sulphate should be reported.

Response: The total number of women 92.6% (n=87) that received magnesium sulfate has included in line 174-176.

“Almost all the participants 96.1% (n=99) received anti-hypertensive medication and 92.6% (n=87) received magnesium sulfate.”

Table 1 p9-10 – add n value for all characteristics where reported data is <103

Response: The n value has been added for all values where data was <103 on page 10 and 11.

---

## [Editor Report · Decision Letter 1]

12 Oct 2020

Risks of adverse perinatal and maternal outcomes among women with hypertensive disorders of pregnancy in southwestern Uganda

PONE-D-20-20782R1

Dear Dr. Lugobe,

We’re pleased to inform you that your manuscript has been judged scientifically suitable for publication and will be formally accepted for publication once it meets all outstanding technical requirements.

Kind regards,

Dell Horey

Academic Editor

PLOS ONE

Additional Editor Comments (optional):

Thank you for the amendments made to this paper. I believe that it is now suitable for publication although I suggest the following minor amendments.

The addition of footnotes to Tables 3 and 4:

Table 3: Adjusted for history of HDP in prior pregnancy, administration of magnesium sulfate and gravidity

Table 4: Adjusted for level of education, mode of delivery and magnesium sulfate administration

Minor edit to the additional statement (line 25-26) in abstract, which is missing a word. It could be written “Among women with HDP at our institution, the majority had preeclampsia with severe symptoms or eclampsia and an unacceptably high rate of adverse perinatal and maternal outcomes; over a fifth of the mothers experienced stillbirth.”
---

## [Editor Report · Acceptance letter]

19 Oct 2020

PONE-D-20-20782R1 

Risks of adverse perinatal and maternal outcomes among women with hypertensive disorders of pregnancy in southwestern Uganda 

Dear Dr. Lugobe:

I'm pleased to inform you that your manuscript has been deemed suitable for publication in PLOS ONE. Congratulations! Your manuscript is now with our production department. 

Kind regards, 

on behalf of

Dr. Dell Elizabeth Horey 

Academic Editor

PLOS ONE